# Impaired Recovery from Influenza A/X-31(H3N2) Infection in Mice with 8-Lipoxygenase Deficiency

**DOI:** 10.3390/medsci7040060

**Published:** 2019-04-12

**Authors:** Rana Alfardan, Changxiong Guo, Linda A. Toth, Daotai Nie

**Affiliations:** 1Department of Medical Microbiology, Immunology, and Cell Biology, Southern Illinois University School of Medicine, Springfield, IL 62794, USA; ranaaziz@stu.edu.iq (R.A.); cjguo@wustl.edu (C.G.); 2Department of Pharmacology, Southern Illinois University School of Medicine, Springfield, IL 62794, USA; toth.linda@frontier.com

**Keywords:** influenza A, X-31, lipoxygenase, *ALOX8*, inflammation, resolution, knockout mice

## Abstract

Lipoxygenase-derived lipid mediators can modulate inflammation and are stimulated in response to influenza infections. We report an effect of 8-lipoxygenase (ALOX8) on the recovery of mice after infection with Influenza virus X31. We compared the responses of 3- and 6-month-old mice with a deletion of *ALOX8 (ALOX8^−/−^)* to influenza infections with those of age-matched littermate wild-type mice (*ALOX8^+/+^*). The duration of illness was similar in 3-month-old *ALOX8^−/−^* and *ALOX8^+/+^* mice. However, the 6-month-old *ALOX8^−/−^* mice showed a prolonged state of illness compared with *ALOX8^+/+^* mice, as evidenced by reduced body temperatures, reduced locomotor activities, and delayed weight recovery. Although residual viral RNA in the lungs at day 10 post-inoculation was significantly influenced by the age of the *ALOX8^−/−^* mice, there were no significant differences between *ALOX8^−/−^* and *ALOX8^+/+^* mice within the same age groups. The levels of cytokines interleukin 6 (IL-6) and keratinocyte chemoattractant (KC) differed significantly between 6-month-old *ALOX8^−/−^* and *ALOX8^+/+^* mice 10 days after viral inoculation. Our data suggest that *ALOX8* deficiency in mice leads to impaired recovery from influenza infection in an age-dependent manner.

## 1. Introduction

Influenza is a transmissible disease caused by a family of RNA viruses. In humans, symptoms of this infection include fevers, headaches, and body aches and pains. In more serious cases, influenza can lead to life-threatening complications such as bronchitis and pneumonia. The viral infection itself and the responding host defense mechanisms can both contribute to these symptoms. These host defense mechanisms include humoral and cellular immune responses involving several cytokines and chemokines. A recent study of lipids in human nasal washes of influenza patients and in bronchoalveolar lavage fluid collected from influenza-infected mice reported that a number of bioactive lipids, including those derived from arachidonic acid metabolism by lipoxygenases, showed marked changes over the course of infection [1].

Lipoxygenases (LOX) are a family of non-heme iron-containing proteins that catalyze the dioxygenation of polyunsaturated fatty acids (PUFAs) containing the 1-*cis*-4-*cis*-pentadiene moiety to form bioactive lipids [2]. Six LOXs have been identified in humans and seven in mice. Metabolism of arachidonic acid by LOXs leads to the formation of regioisomeric *cis/trans* conjugated hydroxyeicosatetraenoic acids (HETEs), leukotrienes, lipoxins, and hepoxilins. Depending on the predominant position of the incorporation of hydroperoxy function, LOXs are classified as 3-, 5-, 8-, 12(S)-, 12(R)-, and 15-LOXs, whose main products are 3(S)-, 5(S)-, 8(S)-, 12(S), 12(R)-, and 15(S)-HETE, respectively [3]. LOX metabolites, including hydroperoxy and hydroxyl fatty acids, have been implicated in a number of cellular processes including proliferation, differentiation, and apoptosis [2].

Some of the bioactive lipids formed through LOX pathways contribute to the induction and resolution of inflammation [1,4]. For example, two lipid mediators, 13(S)-hydroxyoctadecadienoic acid (13(S)-HODE) and 9(S)-hydroxyoctadecadienoic acid 9(S)-HODE, were identified recently as novel biomarkers of inflammation in influenza infection [1]. In addition, the lipid mediator protectin D1 (PD1) shows anti-inflammatory and anti-viral activity by interfering with the replication processes of the influenza virus [4]. The 12- and 15-LOX lipid metabolites, including lipoxins, 12(S)-HETE, 15(S)-HETE, and hepoxilin, can influence the anti-inflammatory response and the resolution of inflammation after infection [5]. For example, lipoxin A4 can block the migration of the neutrophils to a site of the infection and reduce the inflammation [6]. In addition, lipoxins contribute to the resolution of inflammation by activating the phagocytosis of dead cells by monocytes and macrophages [7].

During the course of sub-lethal infections of mice with influenza viruses PR8 and X31, the level of 8(S)-HETE, the main metabolite of ALOX8, was markedly increased in bronchoalveolar lavage fluids at day 5 after influenza inoculation, concurrent with an increase in the expression of ALOX8 in the lung tissues [1]. However, the role of ALOX8 in the progression or resolution of influenza infection is unknown. Here we compared the responses of *ALOX8* knockout mice (*ALOX8^−/−^*) and wild-type controls (*ALOX8^+/+^*) to influenza infection in two age groups. We used clinical parameters such as body temperature, activity and body weight as indicators to the illness or recovery [8]. The data suggest an age-related role for ALOX8 in the recovery of mice after influenza infection. 

## 2. Materials and Methods 

### 2.1. Animals and Characterization of ALOX8 Gene Knockout

All animal studies, including breeding and experimental procedures, were approved by the Laboratory Animal Use and Care Committee of Southern Illinois University School of Medicine in Springfield, Illinois, USA (No. 168-13-025 and 195-10-014). 

*ALOX8* knockout mice (C57BL/6N-ALOX8^tm1a(KOMP)Wtsi^/MbpMmucd) were procured through the University of California Davis Knockout Mouse Project (KOMP) Repository and were bred in-house. 31 *ALOX8^+/+^*, 44 *ALOX8^−/−^*, and 3 ALOX8^+/−^ male mice were used in the experiments. Behavioral experiments were run up to 4 times with 8 to 10 animals. Mice were housed in static micro-isolator cages bedded with Beta chip (Northeastern Products Corporation, Warrenburg, NY, USA) to a depth of an inch. Cages were placed in an environmentally-controlled chamber that was maintained at a controlled temperature (22 ± 1°C) and 12:12 h light: dark cycles under pathogen-free conditions. Cages were changed weekly or more frequently if necessary. Mice were maintained on Purina Lab Diet 5001 (LabDiet, St Louis, MO, USA) and tap water. 

Genomic DNA was extracted from tail or ear biopsies and subjected to polymerase chain reaction (PCR) to determine the genotypes. To determine *ALOX8* gene expression, RNA was extracted from the lungs, reverse transcribed to cDNA, and subjected to quantitative PCR (qPCR) to measure ALOX8 mRNA levels. To determine ALOX8 protein levels, protein lysates obtained from the lung (80 µg per sample) were loaded onto sodium dodecyl sulfate-polyacrylamide gel electrophoresis (SDS-PAGE) gels and transferred to Immobilon-FL membranes from Thermo Fisher Scientific (Cat. No. IPL00010) (Waltham, MA, USA). The membranes were incubated with a 15-LO2 (D-9) antibody from Santa Cruz Biotechnology (Cat. No. sc-271290) (Dallas, TX, USA) at a dilution of 1:250 for one hour at room temperature. This antibody recognizes the internal region for 15-LOX-2 of human and 8-LOX of mouse origin. GAPDH antibody from Santa Cruz Biotechnology (Cat. No. sc-47724) at a dilution of 1:500 was used as a loading control. Goat anti-mouse or rabbit secondary antibodies from LI-COR Biosciences (Cat. No. 92532210) (Lincoln, NE, USA) were used at a dilution of 1:5000 for one hour at room temperature before the membranes were washed and scanned using Odyssey infrared imaging system (LI-COR Biosciences).

### 2.2. Influenza Virus 

Influenza virus strain *A/Hong Kong/X31*(H3N2) was used in these experiments. Viral stocks were prepared by inoculation of embryonated chicken eggs, and the resultant virus-infected allantoic fluid was kindly provided by Dr. Robert Webster (St. Jude Children’s Research Hospital). The titer of the stock was determined by using a 50% tissue culture infective dose (TCID_50_) assay in Madin-Darby Canine Kidney (MDCK) cells. Aliquots of the allantoic fluid were frozen at −80 °C until use and, if necessary, were diluted with PBS prior to use to achieve the desired dose. 

To create the influenza infection in this study, mice were lightly anesthetized with isoflurane and inoculated intranasally. An aliquot of the viral stock was thawed and then diluted with sterile PBS to provide an infectious dose of 7 × 10^^4.65^ plaque-forming units (PFUs)/mL infectious viruses in a volume of 25 μL.

### 2.3. Surgery, Viral Inoculation, and Monitoring of the Infected Mice

One of the studies reported here used intraperitoneal telemetric transmitters to continuously measure the temperature and locomotor activity of mice. Ibuprofen (20 mg/mL) was provided as an analgesic in the drinking water beginning three days prior to the surgery and continuing until day five after the surgery. During surgery, mice were maintained on isoflurane anesthesia and under a heat lamp. The eyes were protected with a bland ophthalmic ointment to retard dryness. A telemetric transmitter was implanted intraperitoneally by using sterile instruments and aseptic technique. During the implantation, mice received 1 mL saline intraperitoneally to maintain the hydration and lubrication of the transmitter and abdominal cavity. Next day, another 1 mL saline was administered intraperitoneally [9].

Mice were housed in groups of five before surgery and individually after surgery. Baseline data were measured at 10 days after surgery and two days before inoculation with the Influenza virus, and continuously thereafter until the end of the experiment. Temperature and locomotor activity were collected continuously from the implanted transmitters, and the average values over the day were reported [10]. Body weights were measured daily. Temperature and activity signals from each individual mouse were obtained through physioTel^TM^ receivers (Data Scientific International, St. Paul, MN, USA) located beneath the animal cage. Dataquest A.R.T.22 software was used to process the data from the data exchange matrix (Data Scientific International, St. Paul, MN, USA). Locomotor activity was detected as movement within the cage based on the transmitter location above the receiver and was recorded as the number of events per 10 min [10]. Data collection continued for up to 19 days after inoculation, by which time all mice had almost recovered, died, or been euthanized during the experiment due to a moribund condition (temperature below 28°C, unresponsive to handling). Euthanasia was performed by isoflurane overdose followed by cervical dislocation. 

In another study, IPTT-300 electronic identification transponders (Bio Medic Data System, Inc., Seaford, DE, USA) were subcutaneously implanted in the interscapular region of mice. Mice were housed in a group of four or less after the implantation. These transponders permit remote collection of temperatures using a wand (model # DAS-6007). The baseline temperature and body weight were collected at 10 days after implantation and daily thereafter until day 10 post-inoculation. On day 10, the mice were euthanized by isoflurane overdose. The lungs were collected for cytokine and residual viral RNA measurements and for eosin and hematoxylin staining.

### 2.4. Hematoxylin and Eosin Staining of Lung Sections

Lung tissue samples from 6-month-old infected *ALOX8^+/+^* and *ALOX8^−/−^* mice were fixed in 10% formalin and embedded in paraffin. Seven micrometer sections were stained with eosin and hematoxylin [11]. Slides were read in a blinded manner. Lung histopathology was visually scored based on the extent of inflammation and the overall appearance of lung architecture [12]. Inflammation was scored as the following: 0 for no inflammation, 1 for mild inflammation (10%–25%; scattered neutrophils), 2 for moderate inflammation (26%–50%) and 3 for marked inflammation (>51%; dense infiltrate). Lung architectural pathology was scored as the following: 0 for normal architecture, 1 for low numbers of necrotic alveolar cells and the presence of swollen alveolar walls, and 2 for high numbers of necrotic alveolar cells, desquamation of the alveolar cells, and swollen alveolar walls.

### 2.5. Measurement of Residual Viral RNA

The *M* gene, which encodes for viral matrix and membrane proteins [13], is considered highly conserved and is usually used to diagnose all influenza A subtypes [14]. We used qPCR to evaluate the residual viral content that was not cleared using RNA extracted from the 100 mg lung tissue of infected mice. qPCR was used as a surrogate for replication and spreading of the infectious virus because it detected the viral genome in the dead and living infected cells [15]. To detect *M* gene RNA levels, we used the forward primer (CATGGAATGGCTAAAGACAAGACC) and reverse primer (TGTCCAAAATGCCCTTAATGG). The Eukaryotic Translation Elongation Factor 1 Alpha 1 (*eEF1a1*) gene was used as a normalization control with forward primer (TCCCTGTGGAAATTCGAGAC) and reverse primer (CCAGGGTGTAAGCCAGAAGA) (*n* = 3 for each genotype and age). *M* gene expression is presented as fold changes compared to the internal control (*eEF1a1*), with raw values further normalized against the mean of the 3-month-old *ALOX8^+/+^* mice. Normalization of all the groups to one standard helps to test if the age and genotype influence the viral RNA copies by using ANOVA. Values shown are mean ± standard error of the mean (SEM), with *N* = 3 per group and with each sample analyzed in triplicate. 

### 2.6. Measurement of Cytokines and Chemokines

Lung tissue (200 mg) from each mouse was homogenized with 500 µL of calcium- and magnesium-free Dulbecco’s phosphate-buffered saline (DPBS) containing 1X protease inhibitor. The samples were stored at −80 °C overnight and then centrifuged at 12,000 rpm for 10 min at 4°C. Protein concentrations in the supernatants were determined using the Pierce bicinchoninic acid (BCA) protein assay. All samples were adjusted to 3 mg/mL protein concentration for measurement of cytokines by using Milliplex® Map kit (Cat. No. MCYTOMAG-70K) (EMD Millipore, Burlington, MA, USA). Direct quantification of the cytokine levels was obtained by using a Power Wave HT Microplate Spectrophotometer (BioTek Instruments, Winooski, VT, USA). The cytokines measured in 6-month-old mice were granulocyte-colony stimulating factor (G-CSF), interleukin-1β (IL-1β), IL-6, interferon γ-inducible protein 10 (IP-10) (also known as C-X-C motif chemokine 10, CXCL10), KC (also known as C-X-C motif chemokine 1, CXCL1), monokine induced by gamma interferon (MIG) (also known as C-X-C motif chemokine 9, CXCL9), interleukin-5 (IL-5) and monocyte chemoattractant protein-1 (MCP-1) (also known as chemokine C-C motif ligand 2, CCL2).

### 2.7. Statistical Analysis

All data are presented as mean values ± SEM. *P* values of <0.05 were considered to indicate a statistically significant effect. *P* values were corrected for multiple comparisons if necessary, by using either the Tukey test or Bonferroni corrections, as indicated below.

Temperature, body weight, and, in the transmitter study, activity data were analyzed by using a mixed model ANOVA with between-subject factors of genotype (*ALOX8^+/+^* and *ALOX8^−/−^*) and age (3- and 6-months old) and a within-subject factor of time (baseline and 19 or 10 days of post-inoculation measurements in the transmitter and chip experiments, respectively). Baseline values represented the average values collected on the two days prior to inoculation. A Bonferroni correction was used to adjust for multiple comparisons within the full model. 

For the residual viral RNA measurements, a two-way ANOVA with post-hoc Tukey testing was used to test for significant differences among the four infected groups as a function of age and genotype. 

Cytokine and chemokine data were log-transformed for analysis and presentation to avoid violations of normality and equal variance. Data were analyzed by using a 2-way ANOVA (infection status and genotype) with post hoc Tukey testing. 

Histopathology (inflammation and architectural scores) was evaluated using 2-way ANOVA (infection status and genotype) with post hoc Tukey testing.

## 3. Results

### 3.1. Characterization of ALOX8 Knockout Mice

The open reading frame of *ALOX8* gene has 677 base pairs coding for 76,230 Dalton protein with 78% homology to human *ALOX15B* gene product [16]. To determine the potential role of ALOX8 in host responses to influenza infections, we obtained mice with *ALOX8* gene knocked out, as evidenced by the genotyping (Figure 1A). Mouse lung tissues collected were also evaluated for ALOX8 expression at RNA and protein levels. As shown in Figure 1B, ALOX8 expression at the RNA level was reduced by approximately half in heterozygous *ALOX8^+/−^* mice and was minimal in homozygous *ALOX8^−/−^* mice. ALOX8 expression at protein level was also minimal in the *ALOX8^−/−^* mice as compared with the *ALOX8^+/+^* mice (Figure 1C). A faint band was noticed in the *ALOX8^−/−^* mice which may be due to low endogenous level of ALOX8 or unspecific detection of other isoforms of lipoxygenases.

### 3.2. Responses Of 3- and 6-Month-Old Littermate ALOX8^+/+^ and ALOX8^−/−^ After Inoculation with Influenza Virus as Measured by Intraperitoneal Transmitter

Body temperature and locomotor activities before and after inoculation with influenza virus were collected by the intraperitoneal transmitter. Analysis of baseline temperatures by 2-way ANOVA (between subject factors of age and genotype) revealed no significant differences (*p* = 0.6061). After intranasal inoculation with influenza virus, the 3-month-old *ALOX8^−/−^* and *ALOX8^+/+^* mice both developed significant reductions in body temperature during days one through 10 post-inoculation, with a reduction persisting in the *ALOX8^−/−^* mice through day 11 and with no significant differences between same-day values for *ALOX8^+/+^* and *ALOX8^−/−^* (Figure 2A). In the 6-month-old mice, significant temperature reductions were detected only at day two for *ALOX8^+/+^* mice, whereas temperatures were significantly lower in the *ALOX8^−/−^* mice for the duration of the post-inoculation period (Figure 2B). Mice entering a moribund state accounted for large temperature variations in the *ALOX8^−/−^* mice between days seven through 16. Temperatures of 6-month-old *ALOX8^+/+^* and *ALOX8^−/−^* mice differed significantly from each other on days eight through 12 and 15. An overall mixed model repeated measures ANOVA revealed significant effects of genotype (*p* = 0.0365) and time after inoculation (*p* < 0.0001), with significant interactions of age*time (*p* < 0.0001), genotype*time (*p* = 0.0063) and age*genotype*time (*p* = 0.0061).

Analysis of baseline locomotor activity by 2-way ANOVA (between subject factors of age and genotype) revealed no significant differences among groups (*p* = 0.180). An overall mixed model repeated measures ANOVA revealed a significant effect of time after inoculation (*p* < 0.0001), with a significant interaction of age*time (*p* = 0.0131). Post-hoc analysis revealed that after intranasal inoculation with influenza virus, 3- and 6-month-old mice of both genotypes developed significant reductions in locomotor activity (Figure 2C,D), with the duration of these reductions lasting one to two days longer in *ALOX8^−/−^* mice when compared with *ALOX8^+/+^* mice at both ages. No significant effects were detected with respect to age (*p* = 0.0642) or genotype (*p* = 0.35). 

Analysis of baseline body weights by 2-way ANOVA (between subject factors of age and genotype) revealed significant differences among groups (*p* < 0.0001). Post-hoc analysis revealed significant differences related to age (3- versus 6-month-old *ALOX8^+/+^* mice, *p* = 0.002; 3- versus 6-month-old *ALOX8^−/−^* mice, *p* = 0.0003), with no significant effect of genotype. An overall mixed model repeated measures ANOVA revealed significant effects of age (*p* = 0.0015), genotype (*p* = 0.0481) and time after inoculation (*p* < 0.0001), with a significant interaction of genotype*time (*p* = 0.0014). Post-hoc analysis revealed that after intranasal inoculation with influenza virus, 3- and 6-month-old mice of both genotypes developed significant reductions in body weight (Figure 2E,F), with the duration of these reductions lasting one to two days longer in *ALOX8^−/−^* mice when compared with *ALOX8^+/+^* mice at both ages. Significant differences between 6-month-old *ALOX8^−/−^* and *ALOX8^+/+^* mice were present on days 10 and 11 post-inoculation. No significant effects were detected with respect to age.

### 3.3. Responses of 3- and 6-Month-Old Littermate ALOX8^+/+^ and ALOX8^−/−^ to Inoculation with Influenza Virus, Monitored Using Subcutaneous Chips

To confirm the observations described above, we used a second approach to monitor the temperature responses of 3- and 6-month-old *ALOX8^+/+^* or *ALOX8^−/−^* mice to influenza infection. In this study, IPTT-300 electronic identification transponders were subcutaneously implanted in the interscapular region of mice, thereby avoiding potential adverse effects of invasive abdominal surgeries. Subcutaneous temperature and body weight were measured daily pre-inoculation and for 10 days post-inoculation with the influenza virus. 

For temperature, baseline values were not significantly different among groups. The overall mixed model repeated measures ANOVA revealed significant effects of genotype (*p* = 0.0093) and of time after inoculation (*p* < 0.0001), with significant interactions of age*time (*p* = 0.0128) and genotype*time (*p* < 0.0001). Post-hoc analysis revealed that baseline temperatures of *ALOX8^+/+^* and *ALOX8^−/−^* mice were not significantly different from each other at either age. After intranasal inoculation with influenza virus, the 3-month-old *ALOX8^+/+^* mice developed significant reductions in body temperature during days one through nine after inoculation, whereas in the *ALOX8^−/−^* mice, the reduction persisted through day 10, with no significant differences between the two strains on any day of the study (Figure 3A). In the 6-month-old mice, both genotypes showed significant reductions in temperature on days one through 10 (Figure 3B). Temperatures of 6-month-old *ALOX8^+/+^* and *ALOX8^−/−^* mice differed significantly from each other on post-inoculation days 7, 8 and 10.

For body weight, baseline values differed significantly among groups (*p* = 0.0006), with post-hoc analysis showing that 6-month-old mice of both strains weighing significantly more than 3-month-old mice of the same strain (*ALOX8^+/+^* mice, *p* = 0.0293; *ALOX8^−/−^* mice, *p* = 0.0175). The overall mixed model repeated measures ANOVA revealed significant effects of age (*p* = 0.0007), genotype (*p* = 0.0122), and time after inoculation (*p* < 0.0001), with significant interactions of age*time (p<0.0001), genotype*time (*p* < 0.0001), and age*genotype* time (*p* < 0.0001). Baseline weights of *ALOX8^+/+^* and *ALOX8^−/−^* mice were not significantly different from each other at either age, but both strains showed a significant age-related increase in weight (*p* = 0.0293 and *p* = 0.0175 for *ALOX8^+/+^* and *ALOX8^−/−^* mice, respectively) (Figure 3C,D). All four groups showed significant reductions in body weight during post-inoculation days one through 10. Post-hoc analysis revealed that body weights of 6-month-old *ALOX8^+/+^* and *ALOX8^−/−^* mice differed significantly from each other on post-inoculation days seven through 10.

### 3.4. Residual Viral RNA in 6-Month-Old ALOX8^−/−^ and ALOX8^+/+^ Mice 10 Days After Inoculation

To determine the mechanism for the impaired recovery of 6-month-old *ALOX8^−/−^* mice from influenza A infections as compared to their littermate wild-type mice, we examined whether there are differences in the levels of residual viruses, as result of delayed viral clearance, between the two groups of mice 10 days after inoculation. The *M* gene, which encodes for viral matrix and membrane proteins [13], is considered highly conserved and is often used to diagnose all influenza A subtypes [14]. As shown in Table 1, the 6-month-old *ALOX8^−/−^* infected mice had a significantly higher residual viral RNA than the 3-month-old *ALOX8^−/−^* infected mice, whereas the *ALOX8^+/+^* mice did not show a statistically significant effect of age. However, no statistically significant differences in residual *M* gene RNA levels were detected between *ALOX8^+/+^* and *ALOX8^−/−^* mice at either age (Table 1). The data suggest that the impaired recovery of 6-month-old *ALOX8^−/−^* mice from influenza A infections is not likely caused by impaired viral clearance or increased residual viruses.

### 3.5. Histopathology of Lungs of 6-Month-Old ALOX8^−/−^ and ALOX8^+/+^ Mice 10 Days After Infections

Next, we evaluated lung histology to determine whether the impaired recovery of 6-month-old *ALOX8^−/−^* mice from influenza A infections was due to increased tissue inflammation. Lung tissues were collected from mice 10 days after infections, when most littermate wild-type mice recovered, processed for hematoxylin and eosin staining, and evaluated for inflammatory infiltrates and changes in tissue architectures. As shown in Table 2, with regard to inflammation, 2-way ANOVA revealed significant effects of both infection status (*p* = 0.004) and genotype (*p* = 0.009). Post-hoc Tukey testing revealed that infected *ALOX8^−/−^* mice had significantly higher inflammation scores than *ALOX8^+/+^* mice (*p* = 0.024). Qualitatively, lung sections from the infected *ALOX8^+/+^* mice showed mild peribronchiolar inflammation and almost complete resolution of inflammation, while interstitial and peribronchiolar inflammation were significantly more severe in the infected 6-month-old *ALOX8^−/−^* mice (Figure 4).

Regarding the alveolar architecture, a 2-way ANOVA revealed significant effects of both infection status (*p* = 0.043) and genotype (*p* = 0.014), with no significant interactions (Table 2). Post-hoc analysis using the Tukey test revealed that infected *ALOX8^−/−^* mice had significantly higher architectural pathology scores than both *ALOX8^+/+^* mice (*p* = 0.02) and uninfected *ALOX8^−/−^* mice (*p* = 0.021). Lung sections from the infected *ALOX8^−/−^* mice showed high numbers of necrotic cells, desquamation of the alveolar epithelium, and swollen alveolar walls. These features were absent or comparatively mild in the *ALOX8^+/+^* mice. The data suggest that in 6-month-old *ALOX8^−/−^* mice, even 10 days after influenza A infections, there were significantly increased inflammation found in the lungs when compared with their littermate wild-type controls.

### 3.6. Cytokine and Chemokine Levels in Lung of Infected and Uninfected 6-Month-Old ALOX8^−/−^ and ALOX8^+/+^ Mice

We next measured the levels of various cytokines and chemokines in mouse lungs 10 days after inoculation to determine which ones may account for the increased inflammation or impaired recovery of 6-month-old *ALOX8^−/−^* mice. As shown in Table 3, ANOVA revealed no significant differences for IL-1β or IL-5 with respect to infection status or genotype, with no significant interactions. G-CSF, IL-6, KC, MCP-1 and MIG showed significant effects of infection status (*p* < 0.0001) but not genotype. IP-10 showed significant effects of both genotype (*p* < 0.0001) and infection status (*p* = 0.004), with no significant interactions. 

Post-hoc analysis revealed no significant differences between uninfected *ALOX8^+/+^* and *ALOX8^−/−^* mice for any of the analyses. Significant effects (*p* < 0.01) due to infection status were present for G-CSF, IL-6, IP-10, KC, MCP-1, and MIG for both *ALOX8^+/+^* and *ALOX8^−/−^* mice. Infected *ALOX8^+/+^* and *ALOX8^−/−^* mice showed significant differences with respect to IL-6 (*p* = 0.017) and KC (*p* = 0.024). 

## 4. Discussion

Data from two independent experiments with two different monitoring devices suggest that mice with deletion of the 8-lipoxygenase gene (*ALOX8^−/−^*) have an age-related delay in recovery from influenza viral infection. While the 3-month-old *ALOX8^−/−^* and *ALOX8^+/+^* mice showed no statistically significant differences in any of the tested variables, in 6-month-old mice, the recovery was significantly slower in *ALOX8^−/−^* mice, as evidenced by delays in the recoveries of baseline locomotor activity, weight, and body temperature. Researchers showed that mice infected with influenza developed hypothermia and decreased locomotor activity when compared to their conditions one or two days before inoculation [17,18]. Hypothermia, hypoactivity and increased sleeping hours resulting in reducing the food intake and body weight of the virus-infected mice [19]

The two approaches used in the experiment both showed an obvious difference in the body weight between the 6-month knockout and the wild-type group but at slightly different time course. The differences in the time-course might be due to the different experimental settings like the size and the position of the transmitters, surgical procedures used, and the post-surgical housing conditions which can affect the food intake and recoveries of mice. Nevertheless, the two different monitoring approaches both confirmed the delayed recoveries of 6-month *ALOX8^−/−^* mice when compared to their wild-type controls.

One possibility for the delayed recovery of 6-month-old *ALOX8^−/−^* mice from influenza infection may be related to delayed clearance of residual viruses. However, in this study, *ALOX8^−/−^* and *ALOX8^+/+^* infected mice showed no significant difference in residual viral RNA in either of the two age groups. There were increased levels of residue RNAs in 6-month-old mice compared with the 3-month-old groups in both genotypes. However, only the 6-month-old *ALOX8^−/−^* infected mice showed a significant difference in the residual viral RNA as compared with 3-month-old *ALOX8^−/−^* mice. Several studies on mice reported an increase in influenza viral burden and mortality over time [20,21]. For other lipoxygenases, it was reported that ALOX5^−/−^ mice had lower virus titers in the lungs but greater susceptibility to the infection, enhanced immunopathology and decreased pulmonary function [22]. But in our study, the 3-month-old mice of both genotypes had the similar residual viral RNA and recovery patterns, whereas recovery was delayed in the 6-month-old *ALOX8^−/−^* mice when compared with *ALOX8^+/+^* mice despite similar levels of residual viral RNA. Our data suggest that factors, besides residual viruses, are involved in the delayed recovery of 6-month-old *ALOX8^−/−^* mice after infections of influenza viruses.

During immune responses to influenza infections, the innate immune responses are activated at the site of infection, followed by activation of adaptive immune responses. Natural killer (NK) cells enter the location of the infection to eradicate the cells infected with influenza viruses [23]. In addition, monocytes and neutrophils can be recruited rapidly to the infection site and with the assistance of alveolar macrophages, help in viral clearance by engulfing the dead infected cells [24]. Resolution of viral infections is governed by multiple factors. In our studies, although the kinetics of changes in body temperature and body weight are similar in both genotypes in first several days after inoculation, the recovery of 6-month-old mice was delayed in terms of locomotor activities, body weight and temperature. Histochemical analysis revealed the presence of persistent inflammation in the lung tissues of *ALOX8^−/−^* mice at day 10 after viral infections. At this time point, most inflammation was resolved in the wild-type control mice. The data suggest that ALOX8 is involved in the host responses and/or subsequent resolution of inflammation after influenza viral infections.

The progression and resolution of symptoms from influenza infection can be related to the viral load and the elevation of inflammatory cytokines such as IL-6 and IP-10 [25]. In our studies, the levels of G-CSF, IP-10, IL-6, MCP-1, MIG, and KC were increased in the lung tissue extracts from mice infected with the X31 virus when compared with uninfected mice. Loss of body weight can result from the elevated pro-inflammatory cytokines such as KC [26]. Increased G-CSF can lead to the stimulation, proliferation and differentiation of white blood cells [27] and mobilization of hematopoietic stem cells [28]. IP-10 recruits macrophages, lymphocyte T cell, natural killer cell and dendritic cells to the site of the infection [29].

In our study, IL-6 and KC were found to have higher levels in the infected *ALOX8^−/−^* mice than in *ALOX8^+/+^* mice on day 10 after inoculation, which may account for the differences in lung tissue inflammation or recoveries. KC recruits neutrophils to the infected lung [30] while IL-6 is usually involved in the acute phase responses toward infections. Further studies are needed to elucidate how ALOX8 loss leads to the changes in IL-6 and KC levels and whether those changes lead to delayed resolution of inflammations. In addition to cytokines, lipoxygenase metabolism of arachidonic acid or other unsaturated free fatty acids can generate a number of bioactive lipids, some of which have a protective role against influenza-induced immunopathology [22]. These mediators can block the expression of some inflammatory cytokines such as IL-6 and KC [31]. For example, mice with ALOX15, an isoform of ALOX8, knocked out had arthritis, with increased levels of IL-6 and IL-1β as compared with their wild-type controls, and the level of KC was correlated to their body weight loss [26]. More studies are needed to determine whether the bioactive lipid products of ALOX8 can modulate inflammatory responses toward influenza infections directly or through modulating the expression of cytokines IL-6 and KC.

An interesting aspect of the data is the relationship of the findings to age. Three-month-old mice are the biological equivalent of teenagers and college freshmen, technically post-pubertal but not yet mature adults and still undergoing maturation [32]. Differences between 3- and 6-month-old mice could be relevant to factors such as thymic involution, which is on-going in that age range [33]. In our study, 3-month-old *ALOX8^−/−^* mice recovered as quickly as *ALOX8^+/+^* mice, yet the 6-month-old mice did not. On the other hand, *ALOX8^−/−^* mice infected at 6 months of age had a higher residual viral load than did *ALOX8^−/−^* mice infected at 3 months of age. The human orthologue of mouse *ALOX8* is 15-lipoxygenase-2 (*ALOX15B*), which is a senescent gene, with its expression increased when prostate epithelial cells become senescent [34]. However, the exact role of ALOX8 in aging is unknown. Further studies are needed to elucidate the mechanism for age-dependent functions of ALOX8.

In summary, loss of the *ALOX8* gene delays mouse recovery from influenza viral infections in an age-related manner. The delay in recovery is accompanied by changes in lung cytokine levels, possibly leading to persistent inflammation in the lung. Our studies demonstrate a functional role for ALOX8 in host recovery from influenza infection. 

## Figures and Tables

**Figure 1 medsci-07-00060-f001:**
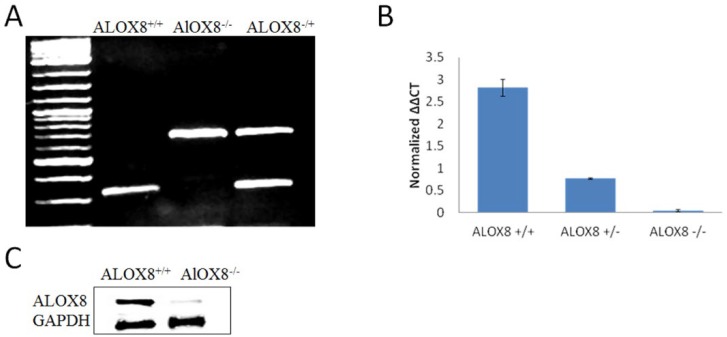
Characterization of *ALOX8^−/−^* mice. All data shown are from uninfected mice. (**A**) Representative genotyping of *ALOX8^+/+^*, *ALOX8^−/−^* and *ALOX8^−/−^* mice. (**B**) qPCR analysis of ALOX8 mRNA levels in the lung of *ALOX8^+/+^*, *ALOX8^+/−^* and *ALOX8^−/−^* mice. (**C**) Western blot analysis of ALOX8 protein levels in the lung of *ALOX8^+/+^* and *ALOX8^−/−^* mice.

**Figure 2 medsci-07-00060-f002:**
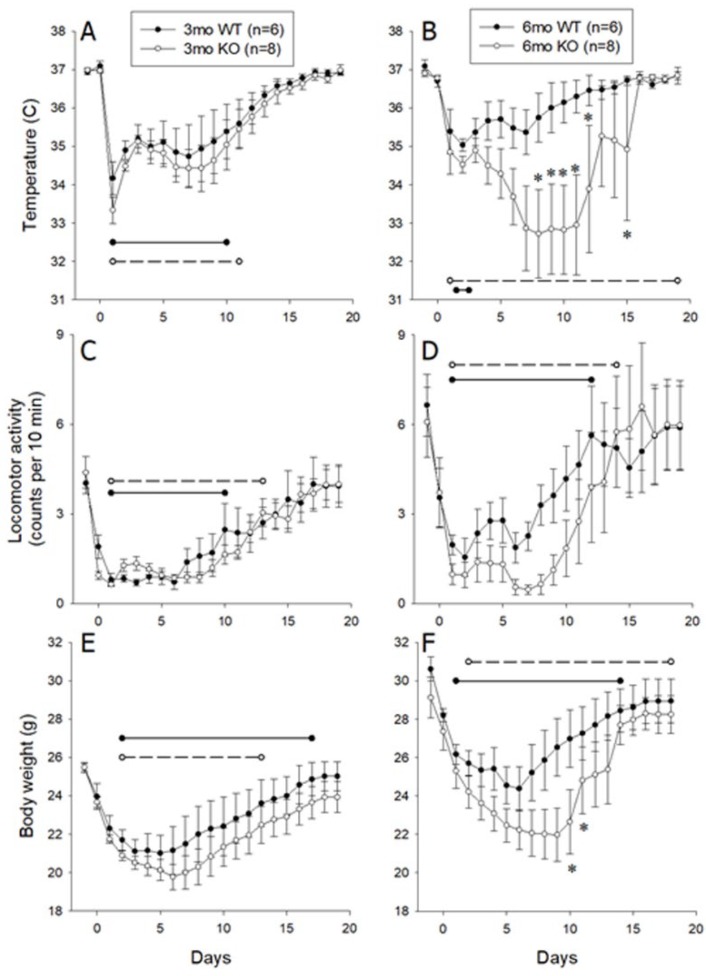
Locomotor activity, temperature and body weight in *ALOX8^+/+^* and *ALOX8^−/−^* mice before and after influenza inoculation. Activity and temperature data were collected using an intraabdominal transmitter. These data are presented as average daily values and SEM based on continuous 10-minute periods of data acquisition for locomotor activity and one reading every 10 minutes for temperature. Body weight was measured daily. Left panels show data from 3-month-old mice (*n* = 6 *ALOX8^+/+^* mice and *n* = 8 *ALOX8^−/−^* mice); right panels show data from 6-month-old mice (*n* = 6 *ALOX8^+/+^* mice and *n* = 8 *ALOX8^−/−^* mice). Horizontal lines denote significant differences from baseline values (*ALOX8^+/+^*, solid; *ALOX8^−/−^*, dashed). Asterisks denote significant difference between *ALOX8^+/+^* and *ALOX8^−/−^* mice at the designated time point.

**Figure 3 medsci-07-00060-f003:**
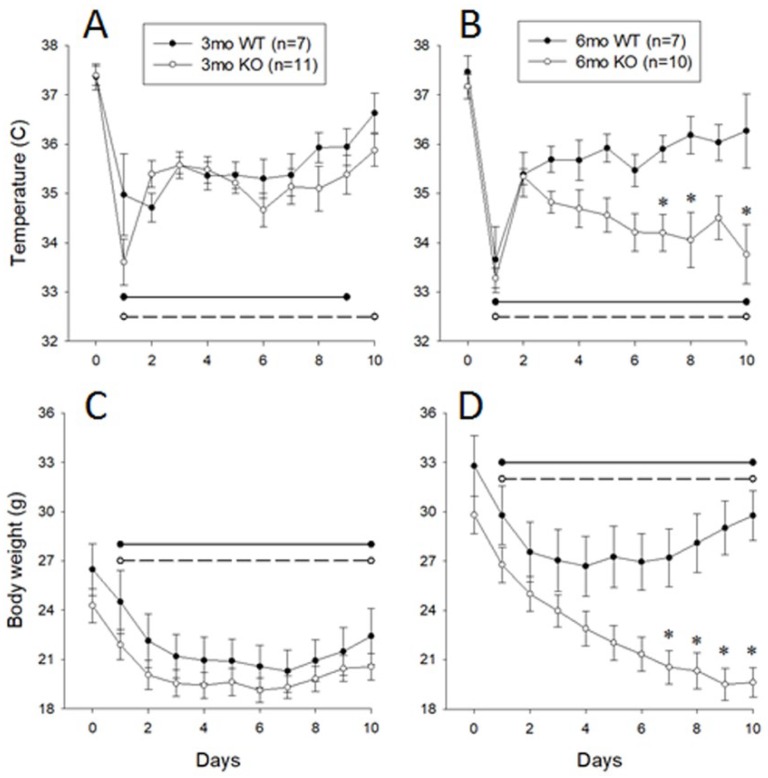
Daily temperature and body weight of *ALOX8^+/+^* and *ALOX8^−/−^* mice before and after influenza inoculation. Mice were implanted with subcutaneous chips to allow remote acquisition of body temperature. Temperature and body weights were measured daily before and for 10 days after influenza inoculation. Data are presented as average daily values ± SEM. Left panels show body temperature (**A**) and weight (**C**) data from 3-month-old mice (*n* = 7 *ALOX8^+/+^* mice and *n* = 11 *ALOX8^−/−^* mice); right panels show body temperature (**B**) and weight (**D**) data from 6-month-old mice (*n* = 7 *ALOX8^+/+^* mice and *n* = 10 *ALOX8^−/−^* mice). Horizontal lines denote significant differences from baseline values (*ALOX8^+/+^*, solid; *ALOX8^−/−^*, dashed). Asterisks denote significant difference between *ALOX8^+/+^* and *ALOX8^−/−^* mice at the designated time point.

**Figure 4 medsci-07-00060-f004:**
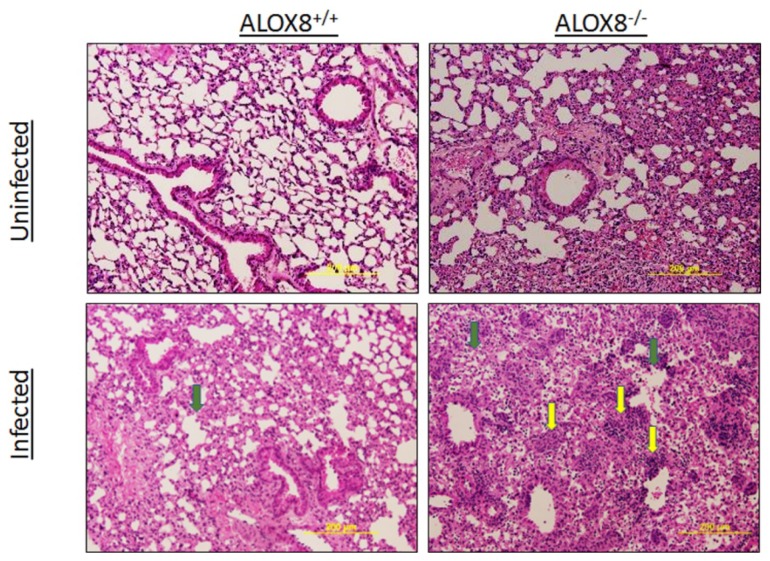
Lung histology changes on day 10 after inoculation of 6-month-old mice. Yellow arrows indicated the inflammation and the green arrows indicated the alveolar pathology changes.

**Table 1 medsci-07-00060-t001:** Residual *M* gene RNA levels in the lungs of influenza-infected mice at 10 days after inoculation. †

Genotype		
Age	*ALOX8^+/+^*	*ALOX8^−/−^*
3 months	0.001 ± 0.000	0.002 ± 0.001
6 months	0.017 ± 0.008	0.018 ± 0.002 *

† *M* gene expression is presented as fold changes compared with the internal control (*eEF1a1*), with raw values further normalized against the mean of the 3-month-old *ALOX8^+/+^* mice. Values shown are mean ± SEM, with *n* = 3 per group. * Post-hoc analysis revealed a significant effect of age for *ALOX8^−/−^* mice but not for *ALOX8^+/+^* mice.

**Table 2 medsci-07-00060-t002:** Histopathology in lungs of infected and uninfected *ALOX8^+/+^* and *ALOX8^−/−^* mice (10 days after inoculation).

	*ALOX8^+/+^*	*ALOX8^−/−^*
**Inflammation Score**		
Uninfected	0 ± 0 (*n* = 3)	0.6667 ± 0.57735 (*n* = 3)
Infected	0.8333 ± 0.75277 (*n* = 6)	2.375 ± 0.91613 (*n* = 8)*
**Architectural Pathology Score**		
Uninfected	0 ± 0 (*n* = 3)	0.3333 ± 0.57735 (*n* = 3)
Infected	0.50 ± 0.54722 (*n* = 6)	1.50 ± 0.75593 (*n* = 8) * †

Data are presented as mean ± STD. * significant difference from uninfected mice of the same genotype; † significant difference between infected *ALOX8^+/+^* and *ALOX8^−/−^* mice.

**Table 3 medsci-07-00060-t003:** Cytokine and chemokine concentrations in the lung of 6-month-old *ALOX8^+/+^* and *ALOX8^−/−^* mice at 10 days after inoculation with influenza virus.

Cytokines	*ALOX8^+/+^*	*ALOX8^−/−^*
IL-1β		
Uninfected	1.38 ± 0.15	1.36 ± 0.10
Infected	1.26 ± 0.11	1.23 ± 0.09
IL-5		
Uninfected	0.22 ± 0.19	0.14 ± 0.16
Infected	0.14 ± 0.04	0.54 ± 0.07
G-CSF		
Uninfected	0.75 ± 0.05	0.75 ± 0.06
Infected	1.95 ± 0.25 *	1.89 ± 0.10 *
IL-6		
Uninfected	0.82 ± 0.09	0.75 ± 0.03
Infected	1.32 ± 0.16 *	1.73 ± 0.05 * †
IP-10 (CXCL10)		
Uninfected	1.75 ± 0.03	1.84 ± 0.03
Infected	3.09 ± 0.08 *	3.23 ± 0.02 *
KC (CXCL1)		
Uninfected	1.49 ± 0.07	1.50 ± 0.04
Infected	1.78 ± 0.05 *	1.97 ± 0.04 * †
MCP-1 (CCL2)		
Uninfected	1.61 ± 0.19	1.55 ± 0.14
Infected	2.65 ± 0.20 *	2.31 ± 0.06 *
MIG (CXCL9)		
Uninfected	1.73 ± 0.03	1.78 ± 0.04
Infected	2.99 ± 0.05 *	3.07 ± 0.02 *

Data are presented as pg/mL, of lung homogenate; values were analyzed and are presented as log-transformed values to avoid violations in normality and equal variance that were present in the non-transformed data. Data are expressed as mean ± SEM. Numbers of samples in each group were as follows: uninfected *ALOX8^+/+^*, *N* = 3; infected *ALOX8^+/+^*, *N* = 6; uninfected *ALOX8^−/−^*, *N* = 3; uninfected *ALOX8^−/−^*, *N* = 8.* *p* < 0.01 as compared with uninfected mice of the same genotype †, *p* < 0.05 as compared with infected mice of the other genotype.

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
