# Peer review of "Impaired Recovery from Influenza A/X-31(H3N2) Infection in Mice with 8-Lipoxygenase Deficiency"

_medsci, 2019, doi:10.3390/medsci7040060_

Reviewer 1 Report

Manuscript submitted by Al-Fardan et al, entitled as “ Impaired Recovery from Influenza A/X-31(H3N2) Infection in Mice with 8-Lipoxygenase Deficiency” studies effect of lipid metabolism on influenza virus infection. Lipid metabolism had been implicated as an important factor for multitude of viral infections, including HBV, HCV etc. Presented study is based on identification of ALOX8 as a potential biomarker in a previous study (Cell, 2013. 154(1): p. 213-27.) ALOX8 seems to be an interesting marker, however authors failed to correlate it well with virus infection. Study does not enhance our understanding of ALOX8 either as a biomarker or as a factor involved in virus replication. My comments, detailed below, may help authors to improve this manuscript.

Fig. 1B, data should be presented as “% of eEF1a1”, to demonstrate level of ALOX8 mRNA in different mice. Authors should have shown level of 8(S)-HETE, the metabolite of ALOX8, to determine functional significance of this gene knockout.

Fig. 2, a mock treated animal (with PBS only) should be compared. Comparison with baseline temperatures, taken 2 days before inoculation, does not reflect upon changes on a daily basis. Also, authors should describe how a dose of 10^5.65 infectious virus in a volume of 25ul considered sublethal? Did they performed LD50 estimation? Also, in the cited reference (Cell, 2013. 154(1): p. 213-27.) authors suggest that “8-HETE, which is derived from 8-lipoxygenase, was detected at a higher level in PR8sublethal- than X31sublethal-infected animals”. It raises a question, why authors of this study used X31 strain, rather than PR8 strain.

Fig.2, what is the rationale to determine temperature and body weight differences among ALOX WT vs KO mice? Why authors presumed that ALOX will be responsible for these differences? Apparently, knocking ALOX8 might have rendered animals more susceptible for virus infection, therefore all these symptomatic differences are being observed.

Data in Table 1 had been overly interpreted. Authors should have compared viral mRNA level per gram of lung tissues. It would have given a better index of productive infection. Alternatively, Bronchoalveolar Lavage should be analyzed, as performed in the cited reference by authors (Ref. 1), which is the main premise for conducting this study.

Lung histology, in Fig. 4, shows remarkable difference between uninfected WT and KO mice. Authors should have presented an image at a higher magnification, and both of a 3-month old and 6-month old uninfected and infected mouse. That would have been helpful in showing that ALOX8 depletion (KO) induced some changes in the lungs over the age, and those changes are responsible for severe influenza symptoms in KO mice.

Table 3 should be presented as bar diagram, comparing different parameters individually between WT and KO mice. This way, expression levels can be understood better. It would result in 8 different panels.  Further, authors should compare cytokine values for a 3-month old to a 6-month old mouse. This reviewer believe that ALOX8 KO mice had aggravated lung condition (based on histopathology) over the age, thus comparing mice at different age could give a better indication of ALOX8 impact, as authors have concluded that ALOX8 renders an age-related delay in recovery from infection.

Authors need to define what does “recovery from infection” means. What factors were considered as recovery? Data on LD50 estimation could be helpful, showing that LD50 for WT mice was higher than for KO mice indicating that KO mice were more susceptible for infection at lower doses.

Minor Comments:

Line 29 “many several…”

Describe generation of ALOX8 Knockout mice in more detail.

Since M gene is routinely used for identification, a reference for the detection primers should have been cited.

Author Response

Dear Editor:

We would like to thank the reviewer for thoughtful and careful review. We have revised the manuscripts after incorporating the suggestions from the reviewers, and we trust the revisions have made the manuscript better. For simplicity, we detail our responses in the following point-to-point format, with reviewers’ comments italicized followed by our responses.

Answers for reviewer 1

Comments and Suggestions for Authors
Manuscript submitted by Al-Fardan et al, entitled as “ Impaired Recovery from Influenza A/X-31(H3N2) Infection in Mice with 8-Lipoxygenase Deficiency” studies effect of lipid metabolism on influenza virus infection. Lipid metabolism had been implicated as an important factor for multitude of viral infections, including HBV, HCV etc. Presented study is based on identification of ALOX8 as a potential biomarker in a previous study (Cell, 2013. 154(1): p. 213-27.) ALOX8 seems to be an interesting marker, however authors failed to correlate it well with virus infection. Study does not enhance our understanding of ALOX8 either as a biomarker or as a factor involved in virus replication. My comments, detailed below, may help authors to improve this manuscript.

Responses:

The reviewer raised an interesting question regarding ALOX8 as biomarker, per the previous study (Cell, 2013, 154(1): p213-227) [1]. This article described the changes in ALOX8 expression and 8(S)-HETE levels during influenza infection and progression, but it was not known whether ALOX8 or 8(S)-HETE modulates influenza infection, progression, or resolution. Our studies used mice with ALOX8 knocked out to determine the possible functions of ALOX8 in the progression of influenza infections. Our data, for the first time, provided the FUNCTIONAL evidences that ALOX8 plays a role in recovery of mice after influenza infections in an age dependent manner.   

Fig. 1B, data should be presented as “% of eEF1a1”, to demonstrate level of ALOX8 mRNA in different mice. Authors should have shown level of 8(S)-HETE, the metabolite of ALOX8, to determine functional significance of this gene knockout.

Responses:

Thanks for the suggestion from the reviewer. For this characterization of ALOX8 mRNA expression, we did not use eEF1a1 as control as we did during measurement of residual viral RNA (The two experiments were done by two different researchers at different times). Here we used actin as the internal control. The data presented were already normalized with the internal controls.  

For 8(S)-HETE level, the reviewer raised a very good question. Mouse ALOX8 is a homologue of human ALOX15B which produces 15(S)-HETE. With mouse ALOX8 as the enzyme for the formation of 8(S)-HETE, one would expect in mice with ALOX8 knocked out, the levels of 8(S)-HETE would be reduced if not completely abolished. However, without extensive further studies, we do not want to extrapolate our results with ALOX8 to 8(S)-HETE for the following two reasons:

First, currently there is no ELISA (EIA) developed for measurement of 8(S)-HETE. Cayman Chemical Company has EIA kit for 15(S)-HETE, with little cross reactivities to 8(S)-HETE. Therefore, lipidomic studies using sophisticated HPLC-MS-MS methods with chiral colums are needed to quantify 8(S)-HETE and other eicosanoids. Further, ALOX8 knockout may lead to increased formation of other HETEs or prostaglandins partly due to the shunting of arachidonic acid to other lipogeneses or cyclooxygenases. Without comprehensive lipidomic analyses, it will be premature to ascribe the changes in disease progression, observed in mice with ALOX8 knocked out, to 8(S)-HETE or other lipids.  

Second, the functional role of ALOX8 in recovery after influenza infections may or may not involve its arachidonate product, 8(S)-HETE. The immediate arachidonate product of ALOX8 is 8(S)-hydroperoxyeicosatetraenoic acid [8(S)-HPETE], which is a reactive compound and can react with other macromolecules including proteins and nucleic acids. 8(S)-HETE is the stable product of 8(S)-HPETE after reduction reaction. 8(S)-HETE, like 15(S)-HETE, can activate PPARs. Given the above considerations, it is obvious that further extensive studies are needed to determine whether exogenous 8(S)-HETE can improve the recovery of mice after influenza infections, especially those with ALOX8 deficiency.

Fig. 2, a mock treated animal (with PBS only) should be compared. Comparison with baseline temperatures, taken 2 days before inoculation, does not reflect upon changes on a daily basis. Also, authors should describe how a dose of 10^5.65 infectious virus in a volume of 25ul considered sublethal? Did they performed LD50 estimation? Also, in the cited reference (Cell, 2013. 154(1): p. 213-27.) authors suggest that “8-HETE, which is derived from 8-lipoxygenase, was detected at a higher level in PR8sublethal- than X31sublethal-infected animals”. It raises a question, why authors of this study used X31 strain, rather than PR8 strain.

Responses:

Due to variations among individual mice, we used the 2 days data of the SAME animal before influenza intranasal inoculation as a baseline to evaluate the changes in body weight, body temperature, and activities after inoculation[2]. The baseline data obtained from the same mouse were considered accurate as the reference value for that animal [3].

The titer of our viral stock was determined by using a TCID50 assay. In our experience using this virus, 10^5.65 infectious virus in a volume of 25 l is not lethal for mice[2, 4]. X31 strain is less potent than PR8. We did infect the recovered mice from X31 infection with PR8 virus. We did not notice a significant difference between the two groups. While the results we obtained from X31 might be applicable to PR8, further studies are needed.  

Fig.2, what is the rationale to determine temperature and body weight differences among ALOX WT vs KO mice? Why authors presumed that ALOX will be responsible for these differences? Apparently, knocking ALOX8 might have rendered animals more susceptible for virus infection, therefore all these symptomatic differences are being observed.

Responses:

We determined body temperature and weight changes since they are good indicators for mouse responses to influenza infections. Body temperature is quite easy to monitor starting from hours after infection and throughout the infection course[5].  Febrile temperature can combat infection by interfering with the viral replication and shedding, and enhancing immune responses towards the invading virus[6]. Fever demands a high metabolic activity. However, hypothermia conserves the energy needed to maintain the metabolic rate for the infected mice[5]. Virus infected mice develop hypothermia, hypoactivity and increase the sleeping hours resulting in reducing the food intake and body weight[7].

ALOX8 expression was elevated at the late phase of the virus infection detected by increased levels of 8(S)-HETE in the BAL mice[1]. Our studies suggest a potential role for ALOX8 in the resolution of inflammations, especially the two genotypes did not show any difference in their susceptibility to the infection. In our study, both WT and KO animals showed same susceptibility for the virus infection. Animals experienced hypothermia, hypoactivity the next day and eventually body weight loss.

Data in Table 1 had been overly interpreted. Authors should have compared viral mRNA level per gram of lung tissues. It would have given a better index of productive infection. Alternatively, Bronchoalveolar Lavage should be analyzed, as performed in the cited reference by authors (Ref. 1), which is the main premise for conducting this study.

Responses:

The concerns from the reviewer are well taken. We used 100 mg of lung tissue to extract mRNA. We added this information in the Materials and Methods.

Lung histology, in Fig. 4, shows remarkable difference between uninfected WT and KO mice. Authors should have presented an image at a higher magnification, and both of a 3-month old and 6-month old uninfected and infected mouse. That would have been helpful in showing that ALOX8 depletion (KO) induced some changes in the lungs over the age, and those changes are responsible for severe influenza symptoms in KO mice.

Responses:

Regarding the potential changes in the lungs over the age as result of ALOX8 depletion, we did not notice any significant changes in the lung sections from uninfected 3 months old KO mice when compared to those from the 6 months old KO mice. However, in the mice 12 months old mice or older, we did notice an increase in the inflammation in the lung of KO mice when compared to the wild-type littermate controls.

Table 3 should be presented as bar diagram, comparing different parameters individually between WT and KO mice. This way, expression levels can be understood better. It would result in 8 different panels.  Further, authors should compare cytokine values for a 3-month old to a 6-month old mouse. This reviewer believes that ALOX8 KO mice had aggravated lung condition (based on histopathology) over the age, thus comparing mice at different age could give a better indication of ALOX8 impact, as authors have concluded that ALOX8 renders an age-related delay in recovery from infection.

Responses:

The suggestions from the reviewer are well appreciated. Originally we presented the data in bar graphs but Dr. Toth, a co-author who has published a number of articles regarding influenza mouse studies, felt the table format was the right way to go, given the multiple statistical analyses involved. The data show the statistical analyses between non-infected and infected mice within same genotypes or between different genotypes.   

The reviewer raised a very interesting question regarding that ALOX8 KO mice might have aggravated lung condition over the age. We measured the cytokines and the chemokines in the infected 3 months old groups but did not notice any significant difference between the ALOX8 KO and WT mice. We did notice that there were increased inflammations in the lungs of ALOX8 KO mice over 12 months old, but not in 6 months old mice, when compared to age-matched controls. Obviously more studies are needed to determine the possible roles of ALOX8 in lung functions.

Authors need to define what does “recovery from infection” means. What factors were considered as recovery? Data on LD50 estimation could be helpful, showing that LD50 for WT mice was higher than for KO mice indicating that KO mice were more susceptible for infection at lower doses.

Responses:

The recovery from viral infections start with the clearance of viral infected cells and resolution of inflammation [8]. Mice infected with viruses exhibit clinical signs of illness like drop in body temperature, activity and body weight which can be used to predict death or recovery. If a mouse had combination of body temperature less than 34.50C and weight loss more than 0.5 g daily would not recover before the experimental endpoint [9].

We did notice the in 6 month old groups, ALOX8 KO mice had reduced survival when compared to their WT controls. Since death was not the endpoint of our approved protocol, we did not determine the LD50.  To determine LD50, a separate animal protocol, with strong rationale why death is used as endpoint, is needed (Most animal use committees discourage the use of death as endpoints, unless surrogate markers for morbidity are not reliable).

Line 29 “many several…”

Responses: It’s corrected.

Describe generation of ALOX8 Knockout mice in more detail.

Responses:

ALOX8 knockout mice (ALOX8tm1a(KOMP)Wtsi), in background of C57BL6 strain, were procured through the University of California Davis Knockout Mouse Project (KOMP) Repository.

Since M gene is routinely used for identification, a reference for the detection primers should have been cited.

Responses:

We designed the primers using M gene sequence. We also used another pair of primers from previous study. The primer sets worked very well when compared to the published primers for dye-based qPCR purpose based on the specific single band of the final PCR product. So we present the data from our own primers.

We trust our revisions have addressed most, if not all, concerns raised the reviewers. With the revisions incorporating reviewers’ suggestions, we trust the manuscript has been significantly improved and it is suitable for publication.

With best regards,

Daotai

Daotai Nie, Ph.D.

Professor

Department of Medical Microbiology, Immunology, and Cell Biology

Southern Illinois University School of Medicine

Simmons Cancer Institute

PO Box 9626

Springfield, IL 62794-9626

Tel: 217-545-9702

Fax: 217-545-3227

References cited:

1.         Tam, V.C., et al., Lipidomic profiling of influenza infection identifies mediators that induce and resolve inflammation. Cell, 2013. 154(1): p. 213-27.

2.         Toth, L.A., J.E. Rehg, and R.G. Webster, Strain differences in sleep and other pathophysiological sequelae of influenza virus infection in naive and immunized mice. J Neuroimmunol, 1995. 58(1): p. 89-99.

3.         Liberati TA, T.R., Randle M, Barrett S, Toth LA., Cytokine and chemokine responses of lung exposed to surrogate viral and bacterial infections. comparative medicine, 2013. 63(2): p. 114-126.

4.         Trammell RA, T.L., Markers for predicting death as an outcome for mice used in infectious disease research. comparative medicine, 2011. 61(6): p. 492–498.

5.         Hart, B.L., Biological basis of the behavior of sick animals. Neurosci Biobehav Rev, 1988. 12(2): p. 123-37.

6.         Husseini, R.H., et al., Elevation of nasal viral levels by suppression of fever in ferrets infected with influenza viruses of differing virulence. J Infect Dis, 1982. 145(4): p. 520-4.

7.         Olivadoti, M.D., et al., Sleep and fatigue in mice infected with murine gammaherpesvirus 68. Brain Behav Immun, 2011. 25(4): p. 696-705.

8.         Sun, J. and T.J. Braciale, Role of T cell immunity in recovery from influenza virus infection. Curr Opin Virol, 2013. 3(4): p. 425-9.

9.         Hankenson, F.C., et al., Weight loss and reduced body temperature determine humane endpoints in a mouse model of ocular herpesvirus infection. J Am Assoc Lab Anim Sci, 2013. 52(3): p. 277-85.

Reviewer 2 Report

The manuscript by Al-Fardan et al. describes the consequences of an influenza infection in mice with a knock-out (KO) in the Alox8 gene. The mutation results in a delayed recovery in aged but not in young mice. This is one of the few studies in the field that performed a thorough statistical analysis using mixed models with repeated measures for a multi-factorial ANOVA and appropriate post-hoc tests. The manuscript is acceptable for publication after some major and minor revisions.

Aged KO mice showed an extended presence of virus RNA compared to wild type (WT) mice of the same age. At the same time, increased inflammatory responses were observed. Thus, it is not clear whether the pathologies observed (body weight loss, lung damage, lung inflammation) are primarily caused by increased virus levels or by the presumed anti-inflammatory functions of Alox8. For the most part of the manuscript the authors describe this correctly (e.g. line 350 ff). However, at some places (e.g. line 336 ff) they claim that the phenotypes must also relate to this presumed anti-inflammatory activities of Alox8. This is not possible to conclude from their studies, because it may as well only be related to increased viral load. Thus, this issue should always be discussed in the above manner: pathology could be caused by either increased viral load or the missing anti-inflammatory activity of Alox8.

In line 363, one could also extend: resolution of inflammations or viral clearance.

The correct nomenclature for genes and proteins is as follows:

Mouse genes: first letter capital, all others lowercase, italic

Mouse proteins: all letters capital, not italic

Human genes: all capital, italic

Human proteins: all capital, not italic

Generic (function, cellular location, etc.): all letters capital, not italic

Please use official gene names, e.g. Eukaryotic Translation Elongation 131 Factor 1 Alpha 1 gene symbol:

Human EEF1A1, see following site for reference: https://www.genecards.org/

Mouse: Eef1a1, see http://www.informatics.jax.org/marker/

It is not exactly clear which surgical procedure has been performed for mice carrying the telemetric transmitter (line 95). Please specify.

The genetic background of the KO and WT mice should be provided. Please check MGD and provide exact nomenclature for the mouse strain, not only for the KO allele.

Viral load has been measured by RT-PCR. It should be clear that this does not measure infectious particles but viral replication. Some virologists may have reservation with this interpretation. However, I think, that this is valid as a surrogate measurement. I recommend that the authors mention this once in the M&M (that it is used as surrogate for replication and spreading of infectious virus).

How was the identity and purity of the virus stock confirmed? This should be done by PCR/sequencing or by NGS.

I did not see an ethics statement for the animal experiments: experiments have been approved by this IACUC and provide the reference number.

In line 103 it is mentioned that activity measures have been taken. Was this possible and done for individual mice? Please clarify.

The genotype of the Alox8 KO is not described. A detailed description of the mutation should be provided in M&M.

Why was the Alox8 RNA not completely absent and why was there still some protein detectable?

Were the histopathological analyses performed in aged or young mice?

There is a mistake in Table 3. The headline of the first column names a gene. I guess this should be generic, like cytokine or else and IL1-ß in the first line of the table, not the headline.

Legend of table 1: it is enough to say that P values were significant, no need to give a specific number which I found confusing.

They describe two experiments that were performed (Figures 2 and 3). The results are somewhat different (significance of differences between body weights in F and D). They should briefly address this in the discussion.

Author Response

Dear Editor:

We would like to thank the reviewes for thoughtful and careful review. We have revised the manuscripts after incorporating the suggestions from the reviewers, and we trust the revisions have made the manuscript better. For simplicity, we detail our responses in the following point-to-point format, with reviewers’ comments italicized followed by our responses.

Answers to Reviewer 2

Aged KO mice showed an extended presence of virus RNA compared to wild type (WT) mice of the same age. At the same time, increased inflammatory responses were observed. Thus, it is not clear whether the pathologies observed (body weight loss, lung damage, lung inflammation) are primarily caused by increased virus levels or by the presumed anti-inflammatory functions of Alox8. For the most part of the manuscript the authors describe this correctly (e.g. line 350 ff). However, at some places (e.g. line 336 ff) they claim that the phenotypes must also relate to this presumed anti-inflammatory activities of Alox8. This is not possible to conclude from their studies, because it may as well only be related to increased viral load. Thus, this issue should always be discussed in the above manner: pathology could be caused by either increased viral load or the missing anti-inflammatory activity of Alox8.

In line 363, one could also extend: resolution of inflammations or viral clearance.

Responses:

We apologize for the confusions from an inadvertent error in the presentation of the data regarding the residual viral RNAs. Actually, our data suggest that there was no significant difference in residual viral RNA levels between the KO and WT mice of 6 month old groups. No significant difference was noted in residual viral RNA levels between the KO and WT mice of 3 month old groups as well. However, we did notice a significant difference between the two KO groups, but not the two WT groups, from two different age groups which worth to mention it.

Our final results including the histopathology and the cytokine levels focusing on the missing anti-inflammatory role of ALOX8 in the 6 months old KO group compared to the WT.

It is not exactly clear which surgical procedure has been performed for mice carrying the telemetric transmitter (line 95). Please specify.

Responses:

For the surgical procedure, ibuprofen (20mg/ml) was provided as analgesic in the drinking water beginning 3 days prior to the surgery and continuing until day 5 after the surgery. During surgery, mice were maintained on isoflurane anesthesia and under a heat lamp. The eyes were protected with ophthalmic ointment to reduce dryness. A transmitter was implanted intraperitoneally by using sterile instruments and aseptic techniques. During the implantation, mice received 1 mL saline intraperitoneally to maintain the hydration and lubrication of the transmitter and abdominal cavity. Next day, another 1 mL saline was administered intraperitoneally. The signals were obtained through physioTel TM receivers (Data Scientific International) located beneath the animal cages. Dataquest A.R.T.22 software was used to process the data from data exchange matrix (DSI).

The genetic background of the KO and WT mice should be provided. Please check MGD and provide exact nomenclature for the mouse strain, not only for the KO allele.

Responses:

C57BL6 background was used. We updated the relevant materials and methods section.

Viral load has been measured by RT-PCR. It should be clear that this does not measure infectious particles but viral replication. Some virologists may have reservation with this interpretation. However, I think, that this is valid as a surrogate measurement. I recommend that the authors mention this once in the M&M (that it is used as surrogate for replication and spreading of infectious virus).

Responses:

Thanks for the suggestion, a sentence has been added to the M&M. qPCR used as surrogate for replication and spreading of infectious virus because it detected the viral genome in the dead and live infected cells [1]

How was the identity and purity of the virus stock confirmed? This should be done by PCR/sequencing or by NGS.

Responses:

The viral stocks were originally prepared by inoculation of embryonated chicken eggs as previously described [2, 3]. Virus-infected allantoic fluid was harvested, TCID50 determined, and aliquots frozen at −80 °C until use.  

I did not see an ethics statement for the animal experiments: experiments have been approved by this IACUC and provide the reference number.

Responses:

The ethics statement has been added.

In line 103 it is mentioned that activity measures have been taken. Was this possible and done for individual mice? Please clarify.

Responses:

Yes, the original data was collected from each mouse in the experiment. For the mice implanted with transmitters, the signals were obtained through physioTel TM receivers (Data Scientific International) located beneath the animal cages. Dataquest A.R.T.22 software was used to process the data from data exchange matrix (DSI). Locomotor activity was detected as movement within the cage based on the transmitter location above the receiver and was recorded as number of events per 10 min [4],

The genotype of the Alox8 KO is not described. A detailed description of the mutation should be provided in M&M.

Responses:

The C57BL6 genetic background has been added in the Materials and Methods section.

Why was the Alox8 RNA not completely absent and why was there still some protein detectable?

Responses:

The qPCR results showed minimal presence of ALOX8 mRNA in the knockout group while in heterozygous group, the reduction was around 50%. The values shown in the bar graphs were mostly due to the baseline noise of qPCR data (Signal to noise ratio).

In contrast to qPCR which shows almost complete absence of ALOX8 mRNA, Western blot showed there was a faint band at the molecular weight similar to ALOX8 in the samples from ALOX8 knockout mice. This may be due to the minor cross reaction with other lipoxygenases present by the antibody used and/or the sample breaching into the well during electrophoresis.

Were the histopathological analyses performed in aged or young mice?

Responses:

The histopathologicall analyses were performed in the aged mice. Description of the group age has been added to the result.

There is a mistake in Table 3. The headline of the first column names a gene. I guess this should be generic, like cytokine or else and IL1-ß in the first line of the table, not the headline.

Responses:

Our thanks to the reviewer. We corrected the error in the table.  

Legend of table 1: it is enough to say that P values were significant, no need to give a specific number which I found confusing

Answer:

Removed.

They describe two experiments that were performed (Figures 2 and 3). The results are somewhat different (significance of differences between body weights in F and D). They should briefly address this in the discussion

Responses:

The suggestions from the reviewer are well taken. We have added several sentences in the discussion to address this. Basically, the two different monitoring approaches were used to confirm whether the 6-moth knockout mice had delay in recoveries when compared to their wild type litter mate controls. Both experiments showed an obvious difference in the body weight between the 6-month knockout and the wildtype group but at slight different time course. The slight difference in the time-course might be due to different experiment settings like the size and the position of the transmitters, surgical procedures used, or the housing conditions that affect the food intake and recoveries of mice.

We trust our revisions have addressed most, if not all, concerns raised the reviewers. With the revisions incorporating reviewers’ suggestions, we trust the manuscript has been significantly improved and it is suitable for publication.

With best regards,

Daotai

Daotai Nie, Ph.D.

Professor

Department of Medical Microbiology, Immunology, and Cell Biology

Southern Illinois University School of Medicine

Simmons Cancer Institute

PO Box 9626

Springfield, IL 62794-9626

Tel: 217-545-9702

Fax: 217-545-3227

References cited:

1.         Nathamon Ngaosuwankul, P.N., Pisut Komolsiri, Phisanu Pooruk, Kulkanya Chokephaibulkit, Tawee Chotpitayasunondh3, Chariya Sangsajja, Charoen Chuchottaworn, Jeremy Farrar and and P. Puthavathana, Influenza A viral loads in respiratory samples collected from patients infected with pandemic H1N1, seasonal H1N1 and H3N2 viruses. Virology Journal, 2010. 7(1): p. 75.

2.         Toth, L.A., J.E. Rehg, and R.G. Webster, Strain differences in sleep and other pathophysiological sequelae of influenza virus infection in naive and immunized mice. J Neuroimmunol, 1995. 58(1): p. 89-99.

3.         Liberati TA, T.R., Randle M, Barrett S, Toth LA., Cytokine and chemokine responses of lung exposed to surrogate viral and bacterial infections. comparative medicine, 2013. 63(2): p. 114-126.

4.         Toth, R.A.T.a.L.A., Behavioral Perturbation and Sleep in Healthy and Virus-Infected Inbred Mice. comparative medicine, 2014. 64(4): p. 283-292.

Reviewer 3 Report

The authors examined the effects of 8-lipoxygenase (ALOX8) on influenza virus infection in mice. The study is novel but there are many confusions about methods, which must be improved to make it clear to readers.

1. The knockout mice are Alox8f/f. The authors even didn't describe how they crossed these mice with which Cre strain to make Alox8-/-. 

2. The levels of all cytokines and chemokine found in the study is < 4 pg/ml, which are very close to the basal levels. Indeed, there is no significant increase after viral infection. How do the authors explain the data?

3. Table 1. It is not clear which two groups were compared to conduct the statistics. Apparently, there is no difference between Alox8 wild type and knockout.

4. Table 1. It should be confirmed by plaque assays.

Author Response

Dear Reviewer:

We would like to thank you for the time and efforts in evaluating our manuscript. Given the some general comments from all reviewers, we added several leading sentences in the results section for better flow of the manuscript. We also tightened up some areas with typos.

Below is the point-to-point responses to your concerns raised.

1.The knockout mice are Alox8f/f. The authors even didn't describe how they crossed these mice with which Cre strain to make Alox8-/-. 

Responses: KOMP, where we obtained the knockout mice, used CSD Knockout First vector to knock out ALOX8 in whole body initially. In the study described in this manuscript, we obtained ALOX8 KO mice through mating between heterozygous males and females.

The review raised a very interesting question regarding conditional knockout. The CSD Knockout First promoter driven vector used by KOMP enables a conversion to a conditional allele via Flp recombination, which can be used to obtain tissue specific knockout of ALOX8. However, we did not use conditional knockout of ALOX8 in this study. It will be very interesting to see whether lung specific knockout of ALOX8 can recapitulate the impaired recovery from influenza of mice with ALOX8 knocked out in whole body as reported in this manuscript.

2. The levels of all cytokines and chemokine found in the study is < 4 pg/ml, which are very close to the basal levels. Indeed, there is no significant increase after viral infection. How do the authors explain the data?

Responses: The levels of the cytokines and chemokine presented in this manuscript were obtained from mice TEN days after infections. Their levels were expected to be high at the beginning of the infection but, as the mice recovered from infections, they should decline. In the present study, the infected mice with sub-lethal dose of X31 at Day 10 showed very low level (ranged between 7 to 0.5 pg/ml) of IL-1beta, IL-5, G-CSF, IL-6, IP-10, KC, and MCP-1 because most mice already recovered from the infections (Please see Fig 2 and 3).

We measured the levels of cytokines and chemokines to investigate whether they are involved in the impaired recovery of 6 month old ALOX8 knockout mice from influenza infections. Post hoc analyses revealed that there were significant differences between infected ALOX8+/+ and ALOX8-/- mice only in IL-6 (p=0.017) and KC (p=0.024), but not in other cytokines or chemokines. Obviously further studies are needed whether IL-6 and KC play a role in the impaired recovery of 6 month old ALOX-/- mice from influenza.

3. Table 1. It is not clear which two groups were compared to conduct the statistics. Apparently, there is no difference between Alox8 wild type and knockout.

Responses: The reviewer is correct that there is no difference between Alox8 wild type and knockout in terms of residual viral RNA at Day 10 after infections. We compared the infected KO to the infected WT to detect any significant difference between them. The statistical analysis did not show any significant difference between these groups. That is why we think it is the delayed resolution of inflammation, not the viral clearance, which cause the impaired recovery of 6 month old ALOX8-/- mice.

However, the statistical analysis showed a significant difference between the two age groups in the infected KO mice, but not in wild type mice. The residual RNA level was higher in the 6 months old infected KO group than the 3 months old infected KO group. Further studies are needed to determine the significance of this subtle difference.

4. Table 1. It should be confirmed by plaque assays.

Responses: The reviewer has point in terms of plaque assays to measure viral loads. However, in Table 1 we present the levels of RESIDUAL viral RNA at Day 10 after viral infection. At this time point, most mice have recovered from viral infections and the viral loads are expected to be too low to be reliably measured by plaque assays. As evident from the data presented in Table 1, the levels of residual viral RNA, as measured by qPCR of M gene RNA, were pretty low of internal control (eEF1a1). In this manuscript, we are very careful in using the term “RESIDUAL viral RNA”, instead of the term “Viral Load”.

We hope we have addressed your concerns to the best we can. Since the manuscript describes, for the first time, the functional role of ALOX8 in recovery from influenza A infections, we hope you find the findings can be published and shared with other researchers in the field.

Sincerely,

Daotai

Daotai Nie, Ph.D.

Professor

Department of Medical Microbiology, Immunology & Cell Biology

Simmons Cancer Institute, Southern Illinois University School of Medicine

Phone: 217-545-9702;

Fax: 217-545-3227;

E-mail: [email protected].

Round  2

Reviewer 1 Report

Manuscript resubmitted by Al-Fardan et al, does not answer concern raised by this reviewer.

This reviewer consider that a productive infection cycle was not established in the mice at the given dose of 10^5.65 infectious virus. The viral titer was calculated by TCID50, which is not a very effective measurement. A plaque forming unit is a rather better measurement. The dose used in the presented study is too low (https://www.atcc.org/en/Global/FAQs/4/8/Converting_TCID50_to_plaque_forming_units_PFU-124.aspx), to establish a productive viral infection. It does reflect in the absence of any virus mediated changes in the WT mice. In Fig.2, in WT mice no significant difference could be seen in terms of changes induced by viral infection. Authors need to define what period do they consider to be “onset of illness” and “resolution of illness”. Why those periods are not present in WT mice?  Another hint that a productive infection may not have established is in Table 1, where M gene expression is very low, comparatively. A fold change of 0.017 or 0.018 is not significant, compared to the internal control, that too with only 3 mice per group. Therefore, this reviewer is of the opinion that a productive infection was not established and thus interpolating changes to ALOX8 is an overstatement.

Similarly, due to absence of a productive infection, cytokine level did not changed much. Also, influenza virus induce an acute immune response, day 10 is a very late time-point to expect any significant change (even if productive infection is established).

Further, as pointed earlier, body temperature depends on day to day housing condition. Therefore, using day-2 as a baseline is an improper control.

Also, in discussion section, in line 360-362, authors have cited a reference indicating that "mice infected with influenza developed hypothermia and decreased locomotor activity when compared to their conditions 2 days before inoculation.[17]. ". However, on cross examination of mentioned literature, this reviewer could not find any reference to body temperature or locomotor activity.  In their response (rebuttal letter), authors cited 3 different references, but none of them indicate that body temperature and locomotor activity are a good method to measure influenza virus infection. Authors need to revisit this parameter.

Authors also did not provide any significant detail regarding generation of ALOX8 KO mice. How many allele of the gene exist, how many of them were deleted, how they were deleted. Author need to define that deletion of ALOX8 is targeted and specific. In Fig. 1A, what does different bands represent?  Also, no functional data is presented to suggest whether ALOX8 deletion is functional. Also, ALOX8+/- (heterozygous) mice could be a better control, rather than ALOX8+/+ mice, which seems to be normal  C57BL/6 mice (no detail provided). A littermate control would have reflected better. Authors did not discuss why heterozygous mice were not studied?

Authors’ response regarding histopathology does not answer concerns raised by this reviewer. It seems that ALOX8 KO had induced lung pathology (see uninfected lung image, compared to WT mice). However, giving a score zero to uninfected KO mice, similar to WT, is an understatement, thus making KO mice histopathology post infection look worse (higher score) than it actually is. Therefore, authors need to revisit their data.

Overall the revised manuscript does not add any value to the previous manuscript. Authors DID NOT provide convincing evidence that

A.    A productive virus infection had been established.

B.    ALOX8 function was disabled in a KO mice, and changes seen in lung histopathology are not due to loss of ALOX8.

C.   Loss of body weight or body temperature is a good method to interpret progression of influenze virus infection.

Therefore, deriving any conclusion from this study can not be justified.

Author Response

Reviewer#1

Manuscript resubmitted by Al-Fardan et al, does not answer concern raised by this reviewer.

 This reviewer consider that a productive infection cycle was not established in the mice at the given dose of 10^5.65 infectious virus. The viral titer was calculated by TCID50, which is not a very effective measurement. A plaque forming unit is a rather better measurement. The dose used in the presented study is too low (https://www.atcc.org/en/Global/FAQs/4/8/Converting_TCID50_to_plaque_forming_units_PFU-124.aspx), to establish a productive viral infection. It does reflect in the absence of any virus mediated changes in the WT mice. In Fig.2, in WT mice no significant difference could be seen in terms of changes induced by viral infection. Authors need to define what period do they consider to be “onset of illness” and “resolution of illness”. Why those periods are not present in WT mice?  Another hint that a productive infection may not have established is in Table 1, where M gene expression is very low, comparatively. A fold change of 0.017 or 0.018 is not significant, compared to the internal control, that too with only 3 mice per group. Therefore, this reviewer is of the opinion that a productive infection was not established and thus interpolating changes to ALOX8 is an overstatement.

Responses: We see the concerns raised by the reviewer. Weight loss and hypothermia indicate illness, but do not necessarily indicate a productive infection. For example, these changes can be produced by substances such as poly I:C or LPS. However, substances like this would have an effect for only a day or 2. The duration of the changes suggests but does not prove a productive infection. The only way to prove that would be to measure virus, which would require killing mice at a relatively early time to do titers which require more animals and duplications of previous results published. We surmise it was a productive infection based on the duration of illness and consistency with the literature and our past experience [1, 2, 3].

We are confused about the statement It does reflect in the absence of any virus mediated changes in the WT mice. In Fig.2, in WT mice no significant difference could be seen in terms of changes induced by viral infection.” In Figure 2, upon viral inoculations, there were significant drops in body temperatures, locomoter activity, and body weight, when compared to the baseline data before inoculation in animals of both genotypes (Please see the Figure 2 figure legend and texts for more details). Similar changes were also observed in Figure 3.

The data in Table 1 were about RESIDUAL RNA levels at Day 10. The purpose of the data is to determine whether there was increased residual viral RNA in 6 month old ALOX8 knockout mice that may explain the delayed recovery. At Day 10, most viruses would be cleared in normal mice.

Based on literatures, our past experience [1, 2, 3], significant changes in body temperatures, body weight and locomoter activity after viral inoculation and duration of illness, we are confident of productive infections in the experiments.

Similarly, due to absence of a productive infection, cytokine level did not changed much. Also, influenza virus induce an acute immune response, day 10 is a very late time-point to expect any significant change (even if productive infection is established).

Responses:  We agreed with the reviewer that “Also, influenza virus induce an acute immune response, day 10 is a very late time-point to expect any significant change (even if productive infection is established)”. The purpose of measuring cytokines at Day 10 is to determine whether some cytokines are related to the delayed recovery of 6 month old ALOX8 knockout mice. At Day 10, the wildtype mice appeared to be fully recovered based on their body temperature, locomoter activity, and body weight. However, there were significant increases in G-CSF, IL-6, IP-10, KC, CCL2, and MIG in varying extents in infected mice when compared to their respective uninfected controls (Table 3). Among those cytokines, only IL-6 and KC were significantly elevated in the infected ALOX8 knockout mice when compared with their infected littermate wildtype mice (Table 3).  

Further, as pointed earlier, body temperature depends on day to day housing condition. Therefore, using day-2 as a baseline is an improper control.

Responses: The animals were kept in an environmentally controlled chamber (Materials and Methods, Section 2.1.). From our experiences handling the mice, the body temperatures are consistent in healthy mice.

Also, in discussion section, in line 360-362, authors have cited a reference indicating that "mice infected with influenza developed hypothermia and decreased locomotor activity when compared to their conditions 2 days before inoculation.[17]. ". However, on cross examination of mentioned literature, this reviewer could not find any reference to body temperature or locomotor activity.  In their response (rebuttal letter), authors cited 3 different references, but none of them indicate that body temperature and locomotor activity are a good method to measure influenza virus infection. Authors need to revisit this parameter.

Responses: The reference has been updated with one of our previous publications [3] as Reference 18 in the manuscript. We apologize for the oversights.  

Authors also did not provide any significant detail regarding generation of ALOX8 KO mice. How many allele of the gene exist, how many of them were deleted, how they were deleted. Author need to define that deletion of ALOX8 is targeted and specific. In Fig. 1A, what does different bands represent?  Also, no functional data is presented to suggest whether ALOX8 deletion is functional. Also, ALOX8+/- (heterozygous) mice could be a better control, rather than ALOX8+/+ mice, which seems to be normal  C57BL/6 mice (no detail provided). A littermate control would have reflected better. Authors did not discuss why heterozygous mice were not studied?

Responses: The animals were procured from a commercial/public source (See Materials and Methods). The strategy and approach to knock out a gene is described in their website and part of public knowledge. Figure 1A describes a typical genotyping result, with bands indicating the PCR products of genomic DNAs with genotyping primer sets. As evident from the figure, the sizes of amplicons were different between ALOX8 knockout mice and their wildtype littermates (The amplicon was larger in knockout mice due to insertion mutation). Heterozygous mice had amplicons of both sizes.   

We did use littermate wildtype mice as controls in all experiments described in the manuscript.

We did not use heterozygous mice in the experiments since most of them were used as breeders to produce enough animals for experiments.

Authors’ response regarding histopathology does not answer concerns raised by this reviewer. It seems that ALOX8 KO had induced lung pathology (see uninfected lung image, compared to WT mice). However, giving a score zero to uninfected KO mice, similar to WT, is an understatement, thus making KO mice histopathology post infection look worse (higher score) than it actually is. Therefore, authors need to revisit their data.

Responses: We apologize for not adequately answering the concerns raised. Indeed, the reviewer raised an excellent point regarding the score given to the uninfected KO mice. We were perplexed as well since we observed persistent elevated inflammation in the lung in old ALOX8 knockout mice (At aged ALOX8 knockout mice, increased carcinogenesis in the lungs was observed as well when mice were housed in regular room). We re-visited the data and found some errors being made when missing numbers were automatically assigned a score of “0”. We updated the data presented in Table 2. We also revised texts based on the updated data.

 Overall the revised manuscript does not add any value to the previous manuscript. Authors DID NOT provide convincing evidence that

A.    A productive virus infection had been established.

B.    ALOX8 function was disabled in a KO mice, and changes seen in lung histopathology are not due to loss of ALOX8.

C.   Loss of body weight or body temperature is a good method to interpret progression of influenze virus infection.

Therefore, deriving any conclusion from this study can not be justified.

Responses: We apologize that our previous revision did not adequately address the concerns raised. For point A, based on the literature, our experiences and durations of illness, productive virus infections were established in our experiments.

For point B, based on genotyping, qPCR to measure ALOX8 mRNA, and Western blot for ALOX8 protein, ALOX8 was indeed knocked out. The changes in lung histopathology after infections were due to two factors: Infections and ALOX8 knockout, per ANOVA analyses.

For point C, in our past experiences [1, 2, 3], body weight, locomoter activity and body temperature were valid measures for the RECOVERY of mice after viral infections, the focus of the manuscript.  

With this revision, we hope that the reviewer can see the scientific merits of the studies, which for the first time demonstrate the cause and effect relationship between ALOX8 and the recovery of mice from influenza infections.

References:

1.            Toth, R.A.T.a.L.A., Behavioral Perturbation and Sleep in Healthy and Virus-Infected Inbred Mice. comparative medicine, 2014. 64(4): p. 283-292.

2.            Trammell, R.A. and L.A. Toth, Effects of Sleep Fragmentation and Chronic Latent Viral Infection on Behavior and Inflammation in Mice. Comp Med, 2015. 65(3): p. 173-85.

3.            Toth, L.A., J.E. Rehg, and R.G. Webster, Strain differences in sleep and other pathophysiological sequelae of influenza virus infection in naive and immunized mice. J Neuroimmunol, 1995. 58(1): p. 89-99.

Reviewer 2 Report

The authors satisfactorily addressed all comments in their revision. Only a few minor issues remain:

The exact full nomenclature for the mouse KO strain should be used, like:

C57BL/6N-Alox8tm1a(KOMP)Wtsi/MbpMmucd (with correct superscripts that cannot be displayed here). Please confirm that this is the correct strain at https://www.mmrrc.org/catalog/sds.php?mmrrc_id=46575

Also for the legend of Table 3: It is enough to say that p values were significant, no need to provide an exact value.

Identity of virus: Did they sequence the HA and NA segments or all segments to confirm exact virus identity?

Author Response

Reviewer#2

The authors satisfactorily addressed all comments in their revision. Only a few minor issues remain:

The exact full nomenclature for the mouse KO strain should be used, like:

C57BL/6N-Alox8tm1a(KOMP)Wtsi/MbpMmucd (with correct superscripts that cannot be displayed here). Please confirm that this is the correct strain at https://www.mmrrc.org/catalog/sds.php?mmrrc_id=46575

Responses: Our thanks to the reviewer in pointing out the exact full nomenclature. We confirmed and updated the nomenclature.  

Also for the legend of Table 3: It is enough to say that p values were significant, no need to provide an exact value.

Responses: Thank you for the suggestions. We revised as suggested.  

Identity of virus: Did they sequence the HA and NA segments or all segments to confirm exact virus identity?

Responses: We contacted the lab of Dr. Robert Webster and they could not find sequencing information. We were assured of the identity of the viruses. The viral stocks were also used in our published studies [1, 2].  

References:

1.            Toth, R.A.T.a.L.A., Behavioral Perturbation and Sleep in Healthy and Virus-Infected Inbred Mice. comparative medicine, 2014. 64(4): p. 283-292.

2.            Trammell, R.A. and L.A. Toth, Effects of Sleep Fragmentation and Chronic Latent Viral Infection on Behavior and Inflammation in Mice. Comp Med, 2015. 65(3): p. 173-85.

Reviewer 3 Report

No further concerns.

Round  3

Reviewer 1 Report

Thanks for critically addressing concerns raised by this reviewer. 

Author Response

Point-to-point responses to Reviewer#1

Thanks for critically addressing concerns raised by this reviewer. 

Responses: We appreciate your time and efforts in reviewing the manuscript. In the revision, we added several leading sentences in the results section so that the manuscript is made easier to follow. We also tightened up some areas such as typos. The changes are tracked in the word file. Hopefully with the revisions, the findings can be published and shared with the researchers in the field.